# Antiviral Agents against Flavivirus Protease: Prospect and Future Direction

**DOI:** 10.3390/pathogens11030293

**Published:** 2022-02-25

**Authors:** Subodh K. Samrat, Jimin Xu, Zhong Li, Jia Zhou, Hongmin Li

**Affiliations:** 1Department of Pharmacology and Toxicology, College of Pharmacy, The University of Arizona, 1703 E Mabel St, Tucson, AZ 85721, USA; sksamrat@pharmacy.arizona.edu (S.K.S.); zli@pharmacy.arizona.edu (Z.L.); 2Chemical Biology Program, Department of Pharmacology and Toxicology, University of Texas Medical Branch, Galveston, TX 77555, USA; jimxu@utmb.edu (J.X.); jizhou@utmb.edu (J.Z.); 3BIO5 Institute, The University of Arizona, Tucson, AZ 85721, USA

**Keywords:** flavivirus, NS2B-NS3, Zika virus, dengue virus, West Nile virus, inhibitors

## Abstract

Flaviviruses cause a significant amount of mortality and morbidity, especially in regions where they are endemic. A recent example is the outbreak of Zika virus throughout the world. Development of antiviral drugs against different viral targets is as important as the development of vaccines. During viral replication, a single polyprotein precursor (PP) is produced and further cleaved into individual proteins by a viral NS2B-NS3 protease complex together with host proteases. Flavivirus protease is one of the most attractive targets for development of therapeutic antivirals because it is essential for viral PP processing, leading to active viral proteins. In this review, we have summarized recent development in drug discovery targeting the NS2B-NS3 protease of flaviviruses, especially Zika, dengue, and West Nile viruses.

## 1. Introduction

The Flavivirus genus of the Flaviviridae family includes more than 70 related arthropod-borne viruses [1,2,3,4]. The most common and representative members are the dengue virus (DENV) with four serotypes (DENV-1, -2, -3, and -4), Zika (ZIKV), West Nile (WNV), Yellow Fever (YFV), Japanese-encephalitis (JEV), and tick-borne encephalitis (TBEV) viruses [5,6,7]. These are the causative agents for viral hemorrhagic fever and encephalitis in human beings.

Susceptible hosts are normally infected by arthropod vectors carrying flaviviruses [1,8,9,10]. Many flaviviruses are restricted to a particular region, though occasional spillovers of viruses can occur, which pose a significant challenge for international health care. For examples, WNV jumped from the Middle East into the Americas [11,12]; and ZIKV spread from Africa to Southeast Asia, the islands of Polynesia, and later to Brazil in 2015–2016, causing an epidemic [3,13,14].

Specific antiviral therapeutics or vaccines are not available to combat most flavivirus infections, except for YFV, DENV, JEV, Kyasanur forest disease virus, and TBEV [10]. In the case of DENV, vaccine design and development have been significantly impeded because of an antibody-dependent enhancement effect due to the presence of four DENV serotypes, leading to a more severe disease phenotype in subsequent natural infections [10,15]. It remains a challenge for the scientific community to develop a DENV vaccine that simultaneously elicits a balanced tetravalent immunity against all four DENV serotypes. In 2019, the United States Food and Drug Administration (FDA) approved the live-attenuated, tetravalent vaccine, Dengvaxia (from Sanofi Pasteur), but only for individuals living in endemic areas between 9–16 years of age with prior DENV infection [15,16]. These challenges increase the demand to develop therapeutics and vaccines that target different flaviviruses.

In addition to vaccines, antiviral therapeutics are urgently needed to treat infected patients in flavivirus outbreaks to use as prophylaxis [17,18]. In this review, we summarize current strategies and examples to develop potential inhibitors against the flavivirus NS2B-NS3 protease complex to combat flavivirus infections.

## 2. Flavivirus Genome Organization

The Flavivirus genome consists of a positive-sense single-stranded RNA which is ∼11 kb in length, consisting of a single long open reading frame (ORF) flanked by a 5′ untranslated region (UTR) and a 3′ UTR [3,4,19,20]. The viral genome is not polyadenylated but capped as in cellular mRNA. The single ORF encodes a PP that is further processed by proteases from the host and the viral NS2B-NS3 complex. The flavivirus protease is highly conserved and essential for virus replication. The viral PP is processed to three structural proteins (Capsid, pr-Membrane, and Envelope) and seven non-structural (NS) proteins (NS1, NS2A, NS2B, NS3, NS4A, NS4B, and NS5) [4,20]. Three structural proteins form the virus shell, whereas seven NS proteins participate in the membrane-bound replication complex. Among these NS proteins, only NS3 and NS5 bear enzymatic functions [18,21,22] (Figure 1).

## 3. Flavivirus Protease

NS3 carries multiple enzymatic functions, including protease, helicase, and triphosphatase activities [23,24]. The NS3 *N*-terminal domain (amino acids (aa) 1–180) is a trypsin-like serine protease with a classic catalytic triad such as Ser135-His51-Asp75 for DENV-2, JEV, WNV and ZIKV, and Ser138-His53-Asp77 for YFV [3,25,26,27,28,29]. A small hydrophilic proportion of NS2B is required as a co-factor for activating the NS3 protease enzymatic function [30,31,32]. The NS3 *C*-terminus domain (aa 180–618) is a superfamily 2 ATPase/helicase that binds to the 3′ end of transient dsRNA and unwinds it in a 3′→5′ direction [33]. In addition, the *C*-terminal domain of NS3 also has an RNA 5′-triphosphatase activity to cap viral RNA [22,34].

NS2B is an integral membrane protein of 14 kDa in size, with three domains: a central hydrophilic domain flanked by two transmembrane segments. The central hydrophilic region of 47 amino acids (spanning aa 49–96) interacts with the NS3 protease as a co-factor (Figure 1) [32,35]. It has been reported that flavivirus NS3 protein is catalytically inactive as a protease in the absence of NS2B in either linked [36,37,38] or unlinked formats [39,40,41,42,43]. Moreover, the NS3 protease domain expressed in *E. coli* is not very soluble without its co-factor NS2B. Proteases among flaviviruses have significant similarity on both structural and sequence levels [44].

The flavivirus NS2B-NS3 protease is one of the most attractive and validated targets for developing a pan flavivirus treatment. Viral proteases in general are highly promising drug targets. Examples include ten HIV-1 protease inhibitors (PIs) [45,46,47] and two HCV PIs [48,49,50]. Thus, it is plausible that a protease inhibitor for flaviviruses will be efficacious in the clinic. This is also evident by the fact that the NS2B-NS3 proteases from different flaviviruses, such as DENV, ZIKV, and WNV, show a high degree of similarity in their sequences and structures. However, lessons from HIV and HCV drug development suggest that PIs have certain drawbacks. Viral variants resistant to PIs were rapidly developed, because both HIV and HCV cause chronic infections [51]. In contrast, most flaviviruses cause acute infections, and drug resistance may be less of a concern.

The NS3 active site has been the main focus of developing flavivirus protease inhibitors. However, only limited success has been achieved, possibly due to the flat and featureless active site and requirement of a charged substrate to bind with P1 and P2 sites which denote the first and second positions of amino acids outward from the protease cleavage site to the N-terminus of the protease substrate [52]. Many of the active site inhibitors are only effective in biochemical assays and show low cellular antiviral activity and poor bioavailability in vivo [51].

## 4. Structural Insight of NS2B-NS3 Protease

Structural study of NS2B–NS3 led to better screening assays to identify PIs [53,54]. The NS2B and NS3 protein complex adopts two different conformations [14]. An “open” inactive conformation is present for the NS2B C-terminal portion when substrate or an active-site inhibitor is absent (Figure 2A) [14,55]. However, binding of an inhibitor or substrate triggered a “closed” conformation for the C-terminal portion of NS2B [56] (Figure 2B). The sequences of the DENV-2 and DENV-3 proteases in Figure 2A and Figure 2B, respectively, are 68% identical. Binding of NS2B to NS3 is required for NS3 function; mutations that disrupt NS2B-NS3 interactions greatly diminish the proteolytic activity of the complex [14,57,58].

## 5. NS2B-NS3 Protease Inhibitors

The functional significance of NS2B-NS3 protease makes it an important target for drug development against flaviviruses. The structural and functional similarity among the proteases of different flaviviruses further demonstrate their significance. Several studies have reported the development of inhibitors of NS2B-NS3 protease that target either the active site, allosteric and orthosteric sites of the enzyme. Collectively, these inhibitors may be referred to as competitive and non-competitive inhibitors, respectively. Several other reviews have been published on flavivirus protease inhibitors in the past [4,59,60]. In this review, we have reviewed the inhibitors against the NS2B-NS3 protease of ZIKV, DENV, and WNV reported in literature between 2015 and 2021.

### 5.1. Competitive Inhibitors

The competitive inhibitor drugs are known to interact with the NS2B-NS3 protease at the active site and inhibit enzyme activity. Several drugs have been identified that work as competitive inhibitors against the flavivirus NS2B-NS3 protease, as shown in Figure 3 and listed in Table 1. In general, except compound **1**, all other competitive inhibitors were not found to display antiviral efficacy in vivo. Alternatively, further experiments will be required to evaluate the in vivo antiviral efficacy for these compounds.

#### 5.1.1. Repurposed Inhibitors

Yuan et al. performed an in silico screening of more than 8000 drugs, followed by biochemical and cellular assays. The FDA-approved drug **1** (Novobiocin) was found to be the most promising inhibitor for the ZIKV NS2B-NS3 protease [61]. Animal studies suggest that Novobiocin treatment reduces viral load in blood and major organs of mice and increases mouse survival rates [61].

Using a structure-based pharmacophore anchor approach, Pathak et al. showed that the FDA-approved drugs of compounds **2** (Asunaprevir) and **3** (Simeprevir) inhibited the ZIKV protease with IC_50_ values of 6.0 µM and 2.6 µM, respectively [62]. These molecules also showed anti-ZIKV activity with EC_50_ values of 4.7 µM and 0.4 µM, respectively [62]. However, these molecules need further validation in animal models for their anti-ZIKV activity.

#### 5.1.2. Synthetic Inhibitors

Initial efforts were made to develop peptide-based protease inhibitors with or without a serine trap moiety such as CF_3_-ketone, an aldehyde, or a boronic acid that may mimic the substrate to resemble the substrate binding mode of the viral polyprotein to the viral protease and block its activity by forming a stabilized inhibitor-protease intermediate. However, peptides are, in general, unstable in vivo and mostly non-permeable to cross cell membranes, so focus has now shifted to the development of small molecules against flavivirus protease [63].

Rassias et al. synthesized a series of novel N-substituted carbazole-based amidines as ZIKV protease inhibitors, and compound **4** (carbazole derivative) demonstrated its in vitro biochemical and cell-based inhibitory profile against ZIKV with an IC_50_ of 0.52 μM and EC_50_ of 1.25 μM [63].

Using quantitative high throughput screenings (qHTS) for small-molecule protease inhibitors, Abrams et al. [64] found that compounds **5** (MK-591) and **6** (JNJ-40418677) inhibited the ZIKV protease with IC_50_ values of 3.0 μM and 3.9 μM, respectively. Moreover, compounds **5** and **6** showed an ability to inhibit ZIKV in human neuronal stem cells using a ZIKV isolate from the 2013 French Polynesian outbreak with EC_50_ values of 3.1 μM and 3.2 μM, respectively [64].

In another study using different in silico methods, compound **7** (4-CF3-benzyl ether), a pan inhibitor of DENV, WNV, and ZIKV proteases, was found to be an active inhibitor against the TBEV protease [65]. In vitro protease assay confirmed that compound **7** inhibited the TBEV protease with an IC_50_ value of 0.92 μM. Another two analogs of compound **7** named in our manuscript as compounds **8** and **9** show strong TBEV protease inhibition with IC_50_ values of 0.97 μM and 3.72 μM, respectively [65]. The antiviral efficacy of these compounds was not reported.

Interestingly compound **8** has also been described as an inhibitor of the NS2B-NS3 proteases of DENV and WNV by Behnam et al. [68]. Compound **8** showed IC_50_ values of 50 nM and 18 nM against the DENV and WNV proteases, respectively. The inhibitor also displayed significant reduction of DENV and WNV titers in cell-based assays with EC_50_ of 3.4 μM and 15.5 µM, respectively [68].

Several synthetic competitive inhibitors have also been identified to target the active site of the DENV NS2B-NS3 protease and affect the cleavage of substrate polyprotein. Compound **10** (Policresulen) is one of the compounds predicted to interact with Gln106 and Arg133 of the DENV-2 NS2B-NS3 protease and affects its stability, leading to inhibition of its protease activity and virus replication with an IC_50_ of 0.81 μg/mL and an EC_50_ of 8.47 μg/mL, respectively [66].

Another approach for inhibiting the protease activity is to generate synthetic cyclic peptides as competitive inhibitors mimicking the substrate of NS2B-NS3 protease. Takagi et al. synthesized an array of cyclic peptides, showing that proper positioning of arginine and aromatic residues in these cyclic peptides enhanced the DENV-specific antiviral activity at an IC_50_ of 0.95 μM for the most potent compound **11 [67]**. Unfortunately, this compound does not have antiviral activity in cell-based antiviral assays. Incorporation of L-2-naphthylalanine and adjustment of positions of arginine led to the discovery of compound **12**, which showed significant anti-DENV activity with EC_50_ of 2.0 µM, although with slightly reduced protease inhibition of IC_50_ of 1.1 µM [67].

In a recent study, Lin et al. explored cyclic peptides based on aprotinin, a pan-serine protease inhibitor composed of a monomeric globular polypeptide of 58 amino acids [69,74]. Because targeting the P side of the enzyme can affect the human serine protease activity, the binding loop of aprotinin was engineered to identify the optimal hydrophobic amino acids for each of the P’ positions, in which P and P’ are referred to amino acid numbers outward from the protease cleavage site to the N- and C-terminus, respectively. The cyclic peptides (CPs) to target both P and P’ sites of the DENV protease active-site pocket were designed to optimize their sequence and length. The best binding cyclic peptide **13** (**CP7**) had a K_i_ value of 2.9 μM against the DENV-3 protease [69]. However, the antiviral potency of these cyclic peptides was not evaluated.

Several non-peptide molecules with inhibitory effect on the DENV NS2B-NS3 protease have been identified and tested against DENV. Two of the tested compounds, **14** (C30H25NO5 (CID: 54692801)) and **15** (C34H23NO7S2 (CID: 54715399)) were shown to have significant inhibitory effects on the NS2B-NS3 protease with an IC_50_ of 17.46 μM and 9.09 μM, respectively [70]. Assessment of anti-DENV activity revealed that these compounds have moderate antiviral activity with EC_50_ values of 14.9 µM and 11.8 µM, respectively [70].

WNV is another medically important flavivirus which needs NS2B-NS3 protease for its replication. Several inhibitors have been designed to date to block the activity of viral protease. Bastos Lima et al. synthesized a group of novel reversible peptide-hybrid inhibitors based on compound **16** (2,4-thiazolidinedione scaffold) [71], showing low micromolar inhibitory activities against the WNV and DENV proteases, although their antiviral efficacy was not investigated.

Recently, synthesis and application of α-aminoalkyl phosphonates and their peptidyl derivatives as the WNV NS2B-NS3 protease inhibitors have been reported [72]. These compounds, acting as irreversible inhibitors, specifically and exclusively react with the protease active site serine residue, leading to the formation of a slow hydrolyzing covalent protease-inhibitor complex. The most potent compound **17** was observed to have a *K_i_* value of 0.4 μM [72]. The antiviral efficacy of this compound was not reported.

#### 5.1.3. Natural Compound Inhibitors

Several studies have shown that compounds derived from natural sources such as plants or microorganisms can exhibit a broad range medicinal properties. In a study using fluorescence-based screening assays it was shown that a natural compound derived from black tea, named compound **18** (theaflavin-3,3′-digallate, ZP10), acted as a potent ZIKV protease inhibitor with an IC_50_ value of 3 μM. ZP10 inhibited ZIKV replication with an EC_50_ of 7.65 μM [73]. Compound **18** was predicted to directly bind to several critical residues at the proteolytic cavity of the NS2B-NS3 protease and inhibited the polyprotein processing [73].

### 5.2. Non-Competitive Inhibitors

The non-competitive inhibitors are known to interact with NS2B-NS3 protease at a site other than the active site (allosteric and orthosteric site) and modulate the enzyme activity. Compared to the flat and featureless active site, the NS2B-binding interface on the NS3 surface has druggable pockets. Since it has been demonstrated that the NS2B binding is essential for protease function, orthosteric inhibitors targeting the NS2B-NS3 interaction interface have been investigated [14,75]. The novel approach avoided the active site’s featureless nature and the requirement for charged inhibitors at the active site. Several recently identified non-competitive inhibitors are discussed in this review, as also shown in Figure 4 and listed in Table 2. Compared to the active site competitive inhibitors (Table 1), many of these non-competitive orthosteric and allosteric inhibitors (Table 2) showed in vivo antiviral efficacy.

Li et al. developed a split luciferase complementation (SLC)-based protein-protein interaction assay to monitor ZIKV NS2B-NS3 interactions [75]. Li et al. screened a total of 2816 approved and investigational drugs. Several orthosteric inhibitors were identified to abolish the NS2B-NS3 interactions with IC_50_ values below 15 μM. Three of them, including compounds **19** (temoporfin), **20** (niclosamide), and **21** (nitazoxanide), could inhibit the DENV-2 NS2B-NS3 protease activity with an IC_50_ of 1.1 ± 0.1, 12.3 ± 0.6, 15.9 ± 0.9 μM and cytotoxicity CC_50_ of 40.7 ± 0.7, 4.8 ± 1.0, 77 ± 7.2 μM in A549 cells, respectively. These molecules are broad-spectrum antivirals against multiple representative flaviviruses with EC_50_ values in a low micromolar to nanomolar range (Table 2). Temoporfin was further tested in animal models showing that it could inhibit viremia and protect 83% of mice from lethal challenge of ZIKV. Importantly, surviving mice did not show any signs of neurological disorder [75]. In another study, nitazoxanide treatment improved the survival rate of mice from a lethal challenge dose of JEV [77]. Tizoxanide, the active metabolite of nitazoxanide, also showed potent antiviral activity towards DENV-2 and YFV [78]. These three compounds inhibited the binding of the NS2B co-factor to NS3 [75].

In addition to the compounds mentioned above, compound **22** (methylene blue) was found to be an orthosteric non-competitive inhibitor, significantly inhibiting the ZIKV NS2B and NS3 protease activity with IC_50_ values in a micromolar range [79]. Moreover, compound **22** significantly inhibited the growth of multiple ZIKV strains and DENV-2 with low micromolar and nanomolar EC_50_ values in cell-based antiviral assays. Furthermore, compound **22** inhibited viral replication in primary neural and placental cells that are relevant to ZIKV pathogenesis. It also protected 3D mini-brain organoids from ZIKV challenge. Animal model studys confirmed that compound **22** treatment significantly improved the survival rate of mice challenged by a lethal dose of ZIKV [79].

Furthermore, compound **23** (erythrosin B) was found to be a potent orthosteric inhibitor of the NS2B-NS3 proteases of ZIKV and DENV-2 with IC_50_ values of 1.7 μM and 1.9 μM, and EC_50_ values of 0.62 μM and 1.2 μM, respectively [80]. SAR studies suggested that compound **23** derivative compound **24** (JMX0902, or Acid Red 94) was slightly more potent than the compound **23** in inhibition of ZIKV replication, with an EC_50_ of 0.3 µM and a slightly reduced IC_50_ of 2.6 µM [81]. The SAR studies indicated that iodine substitutions at R_1_ and R_3_ positions of the xanthene ring were critical for compound **23**’s biological activities, whereas chlorine substitutions were tolerated on the isobenzofuran ring of compound **23** [81].

Recently, a small library of niclosamide derivatives was screened, leading to identification of a new analog. The compound **25** (JMX0207) showed improved efficacy in inhibition of the molecular interaction between NS3 and NS2B, better inhibition of viral protease function, superior pharmacokinetic properties, and enhanced inhibition of viral replication with EC_50_ values of 0.31 µM and 0.3 µM against DENV-2 and ZIKV, respectively [82]. This compound also reduced ZIKV infection in 3D mini brain organoids and significantly decreased viremia in a ZIKV animal model [82].

In another study, Li et al. developed an SLC-based conformational switch assay [14]. This assay monitors the conformational changes of the NS2B C-terminal portion after binding of an active-site inhibitor to the NS2B-NS3 protease complex. In this assay, the authors’ aim was to identify and characterize inhibitors that allosterically prevented the formation of active conformation by NS2B [14]. From a virtual screening pipeline, twenty-nine compounds were selected for testing in a protease inhibition assay. One of these compounds, namely **26** (NSC135618), significantly inhibited the DENV-2 protease function in vitro, with an IC_50_ value of 1.8 μM. Compound **26** could also abolish the protease conformational change in the SLC-based assay. Later, it was found that compound **26** also inhibited viral replication of DENV, ZIKV, WNV, and YFV with EC_50_ values in a low micromolar range [14].

In a screening of 1200 synthesized compounds targeting histone modifying enzymes including lysine specific demethylase1 (LSD1), Yao et al. found compounds **27** (2,3-bis(4-bromophenyl)-5-(piperidin-4-ylmethoxy) pyrazine) and **28** (2,3-bis(4-(tert-butyl) phenyl)-5-(piperidin-4-ylmethoxy) pyrazine) as novel allosteric non-competitive inhibitors with IC_50_ values of 21.7 and 3.1 μM, respectively, against the ZIKV protease [83]. Through further SAR studies, a series of 2,5,6-trisubstituted pyrazine compounds were discovered as broad-spectrum inhibitors of flavivirus proteases. Among them, compounds **29** with a furan-3-yl group and **30** with a tetrahydropyran-3-yl ring exhibited IC_50_ values of 200 nM and 130 nM, respectively, against the ZIKV protease and potently reduced ZIKV replication with EC_68_ values of 300−600 nM, where EC_68_ represents the compound concentration required to lead to half log order of reduction of viral yield [83]. In addition, structural studies revealed that compound 29 bound to an allosteric pocket of NS3, providing a druggable pocket of the flavivirus protease, in contrast to the featureless active site requiring charged substrate [83]. Further studies confirmed that treatment with compound 29 at a dose of 30 mg/kg for 3 days reduced ZIKV viral loads in plasma and brains by up to 96 and 98% (at 24 h), respectively [83,84]. Compound **29** also prolonged mice survival rates.

Hybrid pharmacophores generated by a combination of two individual pharmacophores with 1,2-Benzisothiazol-3(2H)-one (BIT) as the skeleton have been proven to show synergistic effects in inhibition of the DENV-2 NS2B-NS3 [85]. It was observed that BITs bind to the protease in vicinity of the catalytic triad. The BITs containing aromatic substitutions at the N-atom-like compounds **31** with 2-methyl-4-nitrophenyl, **32** with 2-chlorophenyl, and **33** 2,6-dichlorophenyl showed the highest protease inhibition with an IC_50_ of 2.56 ± 1.03 µM, 2.01 ± 0.98 µM, and 5.28 ± 1.89 µM, respectively. The DENV-2 infection assay showed that two BITs, the compound **32** reduced viral infectivity by 1.4 orders of magnitude and the compound **33** reduced the viral infectivity by about six fold [85].

Impermeability of the competitive inhibitors led the researchers to look for non-competitive inhibitors having better properties in terms of affinity, hydrophilicity, and lipophilic ligand efficiency. Millies et al. developed a series of proline-based enantiomers of previously identified non-competitive inhibitors by replacement of 1-tosylprolinyl fragment from the aromatic ether moiety [86]. The substitution pattern of the (sulfone) amide moiety was also judged for their effect on activity of the Zika virus protease. The most potent inhibitor was the 4-nitrophenyl substituted sulfonamide (*R*)-**34** with IC_50_ value of 0.32 μM. Further replacing the proline moiety with a pipecolinic acid afforded compounds (*R*)-**35** and (*S*)-**35** with IC_50_ values of 0.51 μM and 0.58 μM, respectively [86]. Unfortunately, only compound (*R*)-**34** showed moderate to weak antiviral efficacy against DENV and ZIKV.

## 6. Conclusions and Future Direction

This review reports the flavivirus NS2B-NS3 inhibitors developed in the last six years. The flavivirus NS2B-NS3 protease complex is essential for replication of flaviviruses in infected human cells; therefore, its inhibition can affect viral propagation and limit the disease caused by these viruses. Although the structures and binding properties of the flavivirus proteases are well studied, inhibition of the NS2B-NS3 protease has been a challenging task. In recent years, several competitive and non-competitive inhibitors have been identified as having the ability to abolish the activity of these proteases. Potential inhibitors have also been further investigated for their antiviral activity against DENV, ZIKV, WNV, and other flaviviruses in cell-based assays. However, compounds found to be effective in in vitro studies using the NS2B-NS3 protease assay must be investigated further to ascertain their pharmaceutical potential. Factors such as cytotoxicity, cell permeability, and stability in serum, and pharmacokinetics are significant in determining the in vivo efficacy of potential inhibitory compounds. Moreover, the flavivirus NS2B-NS3 protease is a serine protease similar to other host serine proteases. The risk of adverse effects is very high. Therefore, the protease inhibitors targeting the protease P side could likely affect the activities important for host physiology. Inhibitors that target the protease P’ side or its allosteric and orthosteric sites may be better options to counter the flavivirus infection in human individuals. The pharmacokinetic characteristics such as absorption, distribution, metabolism, and excretion properties of these compounds also need to be investigated in order to bring an effective anti-flavivirus drug from the ‘bench to bedside’. The challenging task of flavivirus-specific treatment can be achieved by intensive efforts and sustained multidisciplinary research. The structural similarity between NS2B-NS3 proteases from different members of the Flavivirus family further indicates that studies can be undertaken for designing anti-pan flavivirus compounds having broad-spectrum effects against important human flaviviruses such as DENV, ZIKV, YFV, WNV, and JEV.

## Figures and Tables

**Figure 1 pathogens-11-00293-f001:**
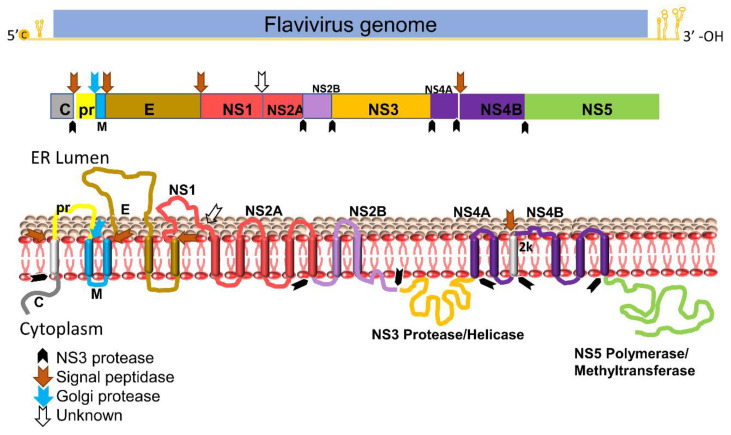
Schematic diagram of flavivirus PP organization, processing, and predicted membrane topology of a mature protein. Top: the representative flavivirus genome. c, RNA cap. Middle: schematic representation of the structural and nonstructural proteins within PP. Black arrows denote cleavage by the viral NS2B-NS3 protease complex, whereas the blue arrow indicates cleavage by the Golgi protease and brown arrows denote cleavage by signal peptidase. White blank arrows indicates unknown protease. Bottom: putative membrane topology of PP predicted by biochemical and cellular analyses, and protease cleavage sites (indicated by arrows).

**Figure 2 pathogens-11-00293-f002:**
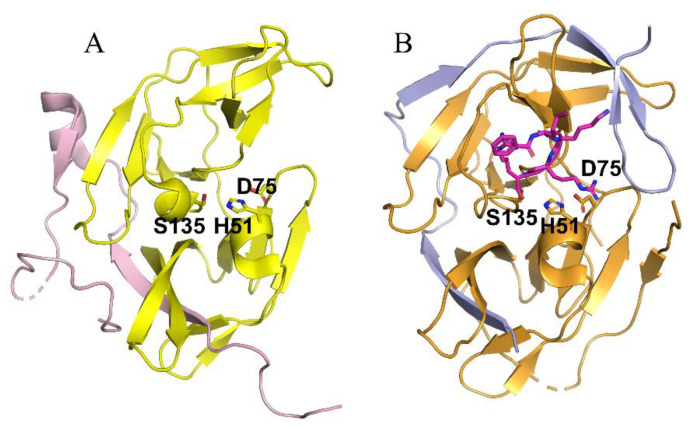
Crystal structures of the NS2B-NS3 proteases of DENV-2 (**A**) and DENV-3 (**B**) in the absence (**A**) and presence (**B**) of a substrate analog. (**A**) The open inactive conformation of the DENV-3 NS2B-NS3 protease in the unbound state (PDB ID of 2FOM). The NS2B cofactor is colored in purple and NS3 protease in yellow. (**B**) The closed active conformation of the DENV-3 NS2B-NS3 protease in complex with a substrate peptide analog (magenta sticks) (PDB ID of 3U1I). The NS2B cofactor is colored in blue and NS3 protease in orange. The catalytic triad His51-Asp75-Ser135 is displayed in stick representation.

**Figure 3 pathogens-11-00293-f003:**
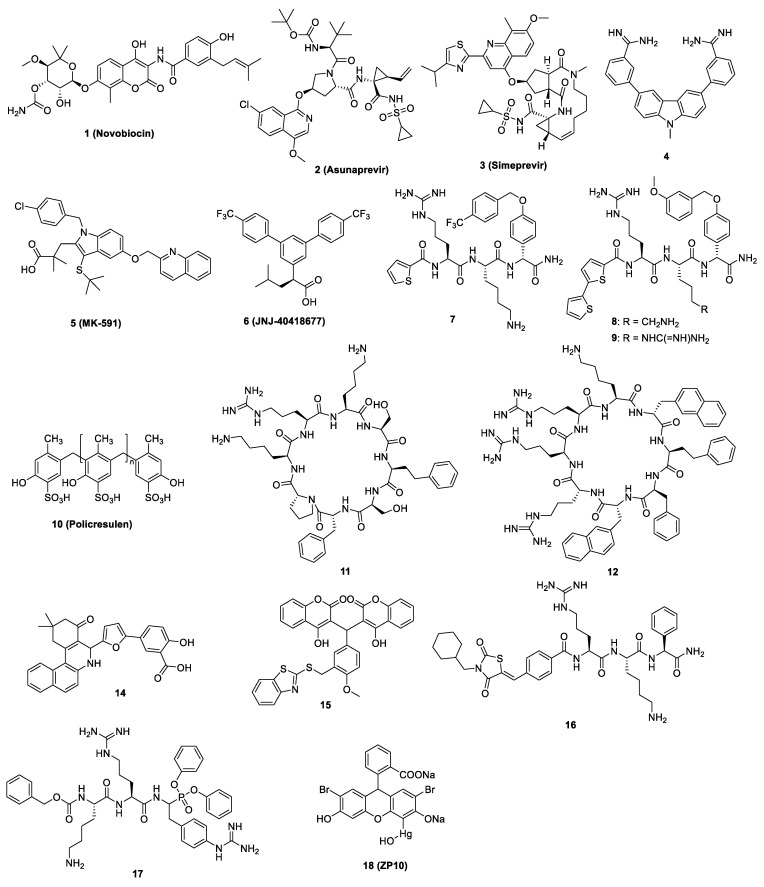
The chemical structures of compounds **1**–**12** and **14**–**18** as competitive inhibitors of the flavivirus NS2B-NS3 protease.

**Figure 4 pathogens-11-00293-f004:**
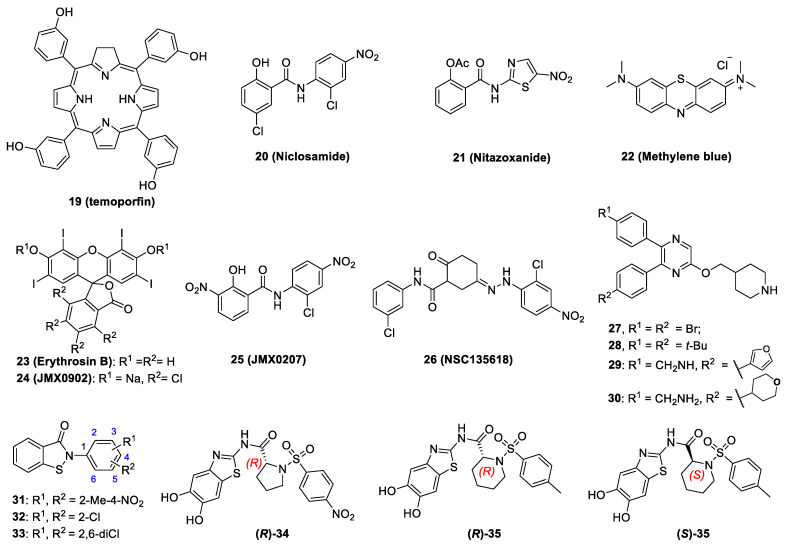
The chemical structures of compounds **19**–**35** as non-competitive inhibitors of flavivirus NS2B-NS3 protease. The positions of the substituents R1 and R2 are depicted with numbers in blue. The configurations of the chiral centers are presented as R and S in red.

**Table 1 pathogens-11-00293-t001:** Competitive inhibitors of flavivirus NS2B-NS3 protease.

Number (Name) of Compound	Targeted Virus	IC_50_ (or K_i_) (μM)	EC_50_ (μM)	In Vivo	Reference
**1** (Novobiocin)	ZIKV	14.2 ± 1.1	42.63	Yes	[61]
**2** (Asunaprevir)	ZIKV	6.0	4.7		[62]
**3** (Simeprevir)	ZIKV	2.6	0.4		[62]
**4** (Carbazole-based amidines)	ZIKV	0.52	1.25		[63]
**5** (MK-591)	ZIKV	3.0	3.1		[64]
**6** (JNJ-40418677)	ZIKV	3.9	3.2		[64]
**7** (4-CF3-benzyl ether)	TBEV	0.92			[65]
	ZIKV	1.64			
**8**	ZIKV	0.25			[65]
	TBEV	0.97			
	DENV	0.05	3.4		
	WNV	0.018	15.5		
**9**	ZIKV	0.94			[65]
	TBEV	3.72			
**10** (Policresulen)	DENV-2	0.81	8.47		[66]
**11**	DENV-2	0.95			[67,68]
**12**	DENV	1.1	2.0		[67]
**13** (PCRARIYGGCA)	DENV-3	Ki = 2.9			[69]
**14** (C30H25NO5)	DENV-2	17.46	14.9		[70]
**15** (C34H23NO7S2)	DENV-2	9.09	11.8		[70]
**16** (Peptide-hybrid inhibitors based on 2,4-thiazolidinedione scaffold)	WNV	0.75			[71]
	DENV	1.05			
**17** (α-aminoalkylphosphonates)	DENV-2	Ki = 0.4			[72]
**18** (Theaflavin-3,3′-digallate (ZP10)	ZIKV	7.65	3		[73]

**Table 2 pathogens-11-00293-t002:** Noncompetitive inhibitors of flavivirus NS2B-NS3 protease.

Number (Name) of Compound	Targeted Virus	IC_50_ (μM)	EC_50_ (μM)	In Vivo	Reference
**19** (Temoporfin)	DENV-2	1.1 ± 0.1	0.020		[75]
	ZIKV		0.024	Yes	
	WNV		0.010		
	JEV		0.011		
	YFV		0.006		
**20** (Niclosamide)	DENV-2	12.3 ± 0.6	0.55		[75,76]
	ZIKV		0.48		
	WNV		0.54		
	JEV		1.02		
	YFV		0.84		
**21** (Nitazoxanide)	DENV-2	15.9 ± 0.9			[75,77]
	ZIKV		1.48		
	JEV		0.39	Yes	
(Tizoxanide)	DENV-2		0.38		[78]
	YFV		0.23		
**22** (Methylene blue)	DENV-2	8.9	0.36		[79] [79]
	ZIKV		0.087–0.2	Yes	
**23** (Erythrosin B)	DENV-2	1.9	1.2		[80]
	ZIKV	1.7	0.62	Yes	
	WNV		0.66		
	JEV		0.35		
	YFV		0.57		
**24** (JMX0902)	ZIKV	2.6	0.3		[81]
**25** (JMX0207)	DENV-2	8.2	0.31		[82]
	ZIKV		0.3	Yes	
**26** (NSC135618)	DENV-2	1.8	0.81		[14]
	ZIKV		1.0		
	WNV		1.27		
	YFV		0.28		
**27**	ZIKV	21.7			[83]
**28**	ZIKV	3.1			[83]
**29**	ZIKV	0.20 ± 0.01	EC_68_ 0.3 or 0.6	Yes	[83,84]
	DENV-2	0.59 ± 0.02			
	DENV-3	0.52 ± 0.06			
	WNV	0.78 ± 0.02			
**30**	ZIKV	0.13	EC_68_ 0.6		[84]
	DENV-2	2.4			
	WNV	0.82			
**31**	DENV-2	2.56 ± 1.03	<30		[85]
**32**	DENV-2	2.01± 0.98	<30		[85]
**33**	DENV-2	5.28 ± 1.89	<30		[85]
(*R*)-**34**	DENV-2	0.32	<3		[86]
(*R*)-**35**	DENV-2	0.51			[86]
(*S*)-**35**	DENV-2	0.58			[86]

## Data Availability

Not applicable.

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
