# Peer review of "Antiviral Agents against Flavivirus Protease: Prospect and Future Direction"

_pathogens, 2022, doi:10.3390/pathogens11030293_

Round 1
Reviewer 1 Report
The review article entitled “Antiviral agents against flavivirus protease: prospect and future direction“ gathers all the recent advances in flavivirus protease drug discovery development. Overall, the manuscript is well written and discuss a highly relevant topic, the development of an antiviral candidate against flavivirus targeting the NS2B-NS3 protease. The proposed manuscript fits on the scope of the journal and the development of an antiviral candidate against flavivirus remains a current concern. Keep in mind that there are several review articles about this same subject published in the past few years, so the review is not unique (although is a good thing author focused in the last 6 years). I suggest add that prior reviews (like https://www.nature.com/articles/nrd.2017.33) cover other time periods. Additionally, there are some points to be reviewed.
3) At the abstract (lines 16-17), authors state that “Flavivirus protease is most attractive target for development of therapeutic antiviral…” In fact, NS3 protease constitute one of the main targets. However, in the state of the art I would disagree with them with that statement. For some viruses, NS5RdRp is the main target (ex. HCV) due the lack of resistance generation (ex. Sofosbuvir).
4) Line 67: Catalytic residues numbers are not the same for all flavivirus proteases. For example, for YFV protease, catalytic triad is Ser138-His53-Asp77 and for DENV2 is Ser135-His51-Asp75.
5) Figure 1: In the membrane topology of the polyprotein, both domains of NS3 (protease and helicase) and NS5 (polymerase and methyltransferase) should be indicated.
6) Line 19: “NS3-NS2B” should be replaced by “NS2B-NS3”
7) Figure 2: It is hard to distinguish the colors in this figure, residue numbering is also difficult to read. For clearness, on figure 1C, superposition should be made using C-alpha trace representation or consider changing the colors.
8) Line 85: “without providing NS2B either in cis or trans.” What the authors mean with cis or trans? NS2B-NS3 protease constructs are usually called linked or unlinked. Cis and trans terminology are usually related to where the cleavage occurs (if it is within or between protein) . see https://www.jbc.org/article/S0021-9258(18)95064-7/fulltext for details
9) Lines 123-124, 234: Terminology, allosteric and orthosteric inhibitors are not the same. – rewrite
10) Lines 136 `an in silco` - in silico
11) lines 276 2.6 µMand – uM and
Author Response
Dear Reviewer, Thank you very much for your constructive comments.
The review article entitled “Antiviral agents against flavivirus protease: prospect and future direction “gathers all the recent advances in flavivirus protease drug discovery development. Overall, the manuscript is well written and discusses a highly relevant topic, the development of an antiviral candidate against flavivirus targeting the NS2B-NS3 protease. The proposed manuscript fits the scope of the journal and the development of an antiviral candidate against flavivirus remains a current concern. Keep in mind that there are several review articles about this same subject published in the past few years, so the review is not unique (although is a good thing author focused on in the last 6 years).
- I suggest adding that prior reviews (like https://www.nature.com/articles/nrd.2017.33) cover other time periods be reviewed.
Response: We have cited several other reviews on NS2B-NS3 protease inhibitors published in the past.
- Boldescu, V.; Behnam, M. A. M.; Vasilakis, N.; Klein, C. D., Broad-spectrum agents for flaviviral infections: dengue, Zika and beyond. Nat Rev Drug Discov 2017, 16 (8), 565-586.
- Nitsche, C., Strategies Towards Protease Inhibitors for Emerging Flaviviruses. Adv Exp Med Biol 2018, 1062, 175-186.
- Brecher, M.; Zhang, J.; Li, H., The flavivirus protease as a target for drug discovery. Virologica Sinica 2013, 28 (6), 326-336.
3) At the abstract (lines 16-17), authors state that “Flavivirusprotease is most attractive target for development of therapeuticantiviral…” In fact, NS3 protease constitute one of the maintargets. However, in the state of the art I would disagree with themwith that statement. For some viruses, NS5RdRp is the main target(ex. HCV) due the lack of resistance generation (ex. Sofosbuvir).
Response: We changed on above line as “Flavivirus protease is one of the most attractive targets for development of therapeutic antivirals because it is essential for viral PP processing, leading to active viral proteins.”
4) Line 67: Catalytic residues numbers are not the same for allflavivirus proteases. For example, for YFV protease, catalytic triadis Ser138-His53-Asp77 and for DENV2 is Ser135-His51-Asp75.
Response: We added catalytic triad of other flaviviruses. “The NS3 N-terminal domain (amino acids (aa) 1-180) is a trypsin-like serine protease with a classic catalytic triad such as Ser135-His51-Asp75 for DENV2, JEV, WNV and ZIKV, and Ser138-His53-Asp77 for YFV.
5) Figure 1: In the membrane topology of the polyprotein, both domains of NS3 (protease and helicase) and NS5 (polymerase and methyltransferase) should be indicated.
Response: We updated the figure 1 with “NS3 protease/helicase and NS5 polymerase/methyltransferase”
6) Line 19: “NS3-NS2B” should be replaced by “NS2B-NS3”.
Response: We have replaced the “NS3-NS2B” by “NS2B-NS3”
7) Figure 2: It is hard to distinguish the colors in this figure, residue numbering is also difficult to read. For clearness, on figure 1C, superposition should be made using C-alpha trace representation or consider changing the colors.
Response: Residue numbering color has been changed for better visibility. Figure 2C has been removed as per reviewer 2 suggestion.
8)Line 85: “without providing NS2B either in cis or trans.” What the authors mean with cis or trans? NS2B-NS3 protease constructs are usually called linked or unlinked. Cis and trans terminology are usually related to where the cleavage occurs (if it is within or between protein) see https://wwwjbcorg/article/S0021
Response: We have changed the sentence as “It has been reported that flavivirus NS3 protein is catalytically inactive as a protease in the absence of NS2B in either linked [37-39] or unlined format [40-44]. Moreover, the NS3 protease domain expressed in E coli is not very soluble without its co-factor NS2B.
9) Lines 123-124, 234: Terminology, allosteric and orthosteric inhibitors are not the same. – rewrite
Response: We changed as follow. “Several studies have reported the development of inhibitors of NS2B-NS3 protease that target either the active site, allosteric and orthosteric site of the enzyme”
10) Lines 136 `an in silco`
Response: in silico Changed to in silico
11) lines 276 2.6 μMand –
Response: uMand changed to μM and
Reviewer 2 Report
Samrat et al. summarise the current literature on reported inhibitors of Flavivirus NS2B-NS3 proteins. The introduction is clear and well written. Section 5 describing the activities of the inhibitors is quite dry and at times seems the authors have copy-pasted text from the cited articles. However, the tables provide a useful summary of the activity of published compounds. I have listed a number of suggested minor amendments, and would also recommend extending the Conclusions/Future Directions as mentioned in the final point below. This would help readers understand more about what is needed for these inhibitors to transition from lab studies, into in vivo, then clinical trial studies.
- Fig 1. Can you relabel the top image? I think ‘genomic polyprotein’ is a misnomer. Do you want to represent the genome or the polyprotein?
- Fig 1. In the bottom image, can you indicate the predicted ER lumen and cytosolic sides of the membrane?
- Sentence beginning Line 88 – rephrase please. Evidence for targeting proteases from unrelated viruses does not demonstrate that NS2B-NS3 are good targets for a pan-flavivirus intervention. It may be more useful to discuss how conserved the sequence of NS2B and NS3 are across flavis, or indicate that several studies have demonstrated inhibitors to be cross-reactive against different flaviviruses.
- Line 99 – please clarify what you mean by P1 and P2 sites
- Fig 2C – I don’t find this part of the figure with overlaid open and closed structures very useful. I recommend removing it. In addition, given DENV-2 serotype is represented in (a) and DENV-3 in (b) it may be useful to mention the identity of the two protein sequences. This would help readers understand how similar you may expect the two structures to be.
- Table 1. There seems to be a few formatting issues with the table. Can you add compound names for 8, 9, 11, 12? If there are not, perhaps refer to as ‘unnamed compound 1’ etc (same for Table 2). If there is a molecular weight available for Policresulen it may be useful to convert EC50 and IC50 values to uM for consistency.
- Line 148 – please clarify what a ‘serine trap’ is.
- Line 167-169 – is [69] the correct reference? I thought their study used ZIKV?
- Line 172 – ‘remains elusive’ could be replaced with ‘were not reported’. Same for line 222.
- Line 189 – do you mean the compound was only active in protease activity assays and not cell based antiviral assays?
- Paragraph beginning line 196 – please define what P and P’ refer to.
- Lines 204, 224 – I’m not familiar with the shorthand viz.?
- Line 289 – it seems like this is a copy-paste from the original article.
- Line 306 – typo for EC50
- A bit more depth in the Discussion/Future directions would be helpful. It is briefly mentioned that the information regarding the pharmacokinetics of many compounds is absent. This could be extended – what properties are required for a compound to be advanced? There are a lot of compounds that have shown antiviral activity in cell-based assays - can the authors also discuss why so many compounds have failed to demonstrate in vivo efficacy? Which compound class is showing the most promise? E.g. are active site inhibitors less likely to advance through development pipelines due to cross reactivity with host protease active site?
Author Response
Dear Reviewer, Thank you very much for your constructive comments
Samrat et al. summarise the current literature on reported inhibitors of Flavivirus NS2B-NS3 proteins. The introduction is clear and well written. Section 5 describing the activities of the inhibitors is quite dry and at times seems the authors have copy-pasted text from the cited articles. However, the tables provide a useful summary of the activity of published compounds. I have listed a number of suggested minor amendments and would also recommend extending the Conclusions/Future Directions as mentioned in the final point below. This would help readers understand more about what is needed for these inhibitors to transition from lab studies, into in vivo, then clinical trial studies.
- Fig 1. Can you relabel the top image? I think ‘genomic polyprotein’is a misnomer. Do you want to represent the genome or thepolyprotein?
Response: We have changed genomic polyprotein to Flavivirus genome
- Fig 1. In the bottom image, can you indicate the predicted ERlumen and cytosolic sides of the membrane?
Response: We have added the indicate the predicted ER lumen and cytosolic sides of the membrane
- Sentence beginning Line 88 – rephrase please. Evidence for targeting proteases from unrelated viruses does not demonstrate that NS2B-NS3 are good targets for a pan-flavivirus intervention. It may be more useful to discuss how conserved the sequence o fNS2B and NS3 are across flavivirus, or indicate that several studies have demonstrated inhibitors to be cross-reactive against different flaviviruses.
Response: We have added the sentence here as” This is also evident by the fact that the NS2B-NS3 proteases from different flaviviruses such as DENV, ZIKV, and WNV show high degree of similarity in their sequences and structures.”
- Line 99 – please clarify what you mean by P1 and P2 sites
Response: We have added and provided references for this. “However, only limited success was obtained, possibly due to flat and featureless active site and requirement of charged substrate to bind with P1 and P2 sites which denote the first and second positions of amino acids outward from the protease cleavage site to the N-terminus of the protease substrate”
- Fig 2C – I don’t find this part of the figure with overlaid open and closed structures very useful. I recommend removing it. In addition, given DENV-2 serotype is represented in (a) and DENV-3 in (b) it may be useful to mention the identity of the two protein sequences. This would help readers understand how similar you may expect the two structures to be.
Response: Figure 2C has been removed. We have mentioned the identity of the two protein sequences. The sequences of the DENV2 and DENV3 proteases in respective Figures 2A and 2B are 68% identical.
- Table 1. There seems to be a few formatting issues with the table.
Can you add compound names for 8, 9, 11, 12? If there are not,perhaps refer to as ‘unnamed compound 1’ etc (same for Table 2).If there is a molecular weight available for Policresulen it may be useful to convert EC50 and IC50 values to uM for consistency.
Response: We thank for the suggestions. Table 1 has been re-formatted to align well. Compounds 8, 9, 11, and 12 are peptide-based compounds and were not given chemical names in original publications; only compound numbers were given in original articles. To keep our manuscript consistent in terms of compound numbering, we believe that sequential numbers for these compounds will be the best strategy not to cause any confusions.
EC50 and IC50 have been recalculated for Policresulen.
- Line 148 – please clarify what a ‘serine trap’ is.
Response: We have added Initial efforts were made to develop peptides-based protease inhibitor with or without a serine trap moiety such as CF3-ketone, an aldehyde or a boronic acid that may mimic the substrate to resemble the substrate binding mode of the viral polyprotein to the viral protease and block its activity by forming a stabilized inhibitor-protease intermediate
- Line 167-169 – is [69] the correct reference? I thought their study used ZIKV?
Response: We have updated the reference. Thank you very much for pointing out the mistake.
- Line 172 – ‘remains elusive’ could be replaced with ‘were not reported’ same for line 222
Response: We have changed remains elusive to was/were not reported.
- Line 189 – do you mean the compound was only active in protease activity assays and not cell based antiviral assays?
Response: Yes, Compound 11. To clarify, we modified the sentence by adding “in cell-based antiviral assay” at the end of the sentence.
- Paragraph beginning line 196 – please define what P and P’ refer to.
Response: We have added and provided references for this at line 99. To be clearer, we added “, in which P and P’ are referred to amino acid numbers outward from the protease cleavage site to the N- and C-terminus, respectively” at the end of the sentence.
- Lines 204, 224 – I’m not familiar with the shorthand viz.?
Response: It has been changed viz to “like” and removed viz from line 224.
- Line 289 – it seems like this is a copy-paste from the original article.
Response: We have rephrased the sentence as “In this assay, authors’ aim was to identify and characterize inhibitors that allosterically prevent the formation of active conformation by NS2B [15]”
- Line 306 – typo for EC50
Response: Thank the reviewer to point out this. However, EC68 was correct as reported in the original literature, which represented the compound concentration required to lead to half log order of reduction of viral yield. To clarify this, we added “ , where EC68 represents the compound concentration required to lead to half log order of reduction of viral yield” at the end of the sentence."
- A bit more depth in the Discussion/Future directions would be helpful. It is briefly mentioned that the information regarding the pharmacokinetics of many compounds is absent. This could be extended – what properties are required for a compound to be advanced? There are a lot of compounds that have shown antiviral activity in cell-based assays - can the authors also discuss why so many compounds have failed to demonstrate in vivo efficacy? Which compound class is showing the most promise? E.g. are active site inhibitors less likely to advance through development pipelines due to cross reactivity with host protease active site?
Response: Conclusion has been changed extensively as per your suggestions.
This review reports the flavivirus NS2B-NS3 inhibitors developed in the last six years. The flavivirus NS2B-NS3 protease complex is essential for replication of flaviviruses in infected human cells; therefore, its inhibition can affect the viral propagation and limit the disease caused by these viruses. Although the structures and binding properties of the flavivirus proteases are well studied, inhibition of the NS2B-NS3 protease has been a challenging task. In recent years, several competitive and non-competitive inhibitors have been identified to have the ability to abolish the activity of these proteases. Potential inhibitors have also been further investigated for their antiviral activity against DENV, ZIKV WNV and other flavivirues in cell-based assays. However, compounds found to be effective in in vitro studies using the NS2B-NS3 protease assay must be investigated further to ascertain their pharmaceutical potential. Factors like cytotoxicity, cell permeability and stability in serum, and pharmacokinetics are significant in determining the in vivo efficacy of potential inhibitory compounds. Moreover, the flavivirus NS2B-NS3 protease is a serine protease similar to other host serine proteases. The risk of adverse effects is very high. Therefore, the protease inhibitors targeting the protease P side could likely affect the activities important for host physiology. Inhibitors which target the protease P’ side or it’s allosteric and orthosteric sites may be better options to counter the flavivirus infection in human individuals. The pharmacokinetic characteristics such as absorption, distribution, metabolism, and excretion properties of these compounds also need to be investigated in order to bring an effective anti-flavivirus drug from the ‘bench to bedside’. The challenging task of flavivirus specific treatment can be achieved by intensive efforts and sustained multidisciplinary research. The structural similarity between NS2B-NS3 proteases from different members of Flavivirus family further indicate that studies can be undertaken for designing anti pan flavivirus compounds having broad-spectrum effects against important human flaviviruses like DENV, ZIKV, YFV, WNV, and JEV.
Reviewer 3 Report
The authors provide a detailed review of different flavivirus protease inhibitors (competitive and non-competitive) that have appeared in the recent literature.
Minor comments:
In Figure 1 I think the legend of the arrows is wrong. If I am not mistaken it should read: the blue arrows indicate cleavage by the Golgi protease and brown arrow denotes cleavage by signal peptidase. Also missing in the legend is the meaning of the white arrow (unknown protease?).
Lines 250-256: Sometimes the reading is difficult. That is, it is difficult to know which viral protease is being discussed. In the original reference (70) it seems that the IC50 data correspond to the DENV-2 protease. However, in Table 2 the data for compound 19 and 20 are on the same line as WNV. In the case of compound 21 there is no DENV-2 included in the table. Summarizing: be careful so that text and table are understood separately and complement each other. The same thing happens again in lines 263-265.
I miss a wider discussion, is too short.
Author Response
Dear Reviewer, Thank you very much for your constructive comments
The authors provide a detailed review of different flavivirus protease inhibitors (competitive and non-competitive) that have appeared in the recent literature.
Minor comments:
- In Figure 1 I think the legend of the arrows is wrong. If I am not mistaken it should read: the blue arrows indicate cleavage by the Golgi protease and brown arrow denotes cleavage by signal peptidase. Also missing in the legend is the meaning of the white arrow (unknown protease?).
Response: We thank the reviewer for pointing out this mistake. We have corrected that and put the legend for white arrow.
- Lines 250-256: Sometimes the reading is difficult. That is, it is difficult to know which viral protease is being discussed. In the original reference (70) it seems that the IC data correspond to the DENV-2 protease. However, in Table 2 the data for compounds 19 and 20 are on the same line as WNV. In the case of compound 21 there is no DENV-2 included in the table.
Summarizing: be careful so that text and table are understood separately and complement each other. The same thing happens again in lines263-265.
Response: We thank the reviewer for pointing out this issue. Due to the formatting of the table, the alignment goes bad. We have corrected that now. DENV-2 is added for compound 21 and its active metabolite Tizoxanide. We also added the following sentence to clarify this. “In another study, nitazoxanide treatment improved the survival rate of mice from a lethal challenge dose of JEV [79]. Tizoxanide, the active metabolite of nitazoxanide, also showed potent antiviral activity towards DENV-2 and YFV [80].”
- I miss a wider discussion, is too short.
Response: We have added some more points in the conclusion.
This review reports the flavivirus NS2B-NS3 inhibitors developed in the last six years. The flavivirus NS2B-NS3 protease complex is essential for replication of flaviviruses in infected human cells; therefore, its inhibition can affect the viral propagation and limit the disease caused by these viruses. Although the structures and binding properties of the flavivirus proteases are well studied, inhibition of the NS2B-NS3 protease has been a challenging task. In recent years, several competitive and non-competitive inhibitors have been identified to have the ability to abolish the activity of these proteases. Potential inhibitors have also been further investigated for their antiviral activity against DENV, ZIKV WNV, and other flaviviruses in cell-based assays. However, compounds found to be effective in in vitro studies using the NS2B-NS3 protease assay must be investigated further to ascertain their pharmaceutical potential. Factors like cytotoxicity, cell permeability and stability in serum, and pharmacokinetics are significant in determining the in vivo efficacy of potential inhibitory compounds. Moreover, the flavivirus NS2B-NS3 protease is a serine protease similar to other host serine proteases. The risk of adverse effects is very high. Therefore, the protease inhibitors targeting the protease P side could likely affect the activities important for host physiology. Inhibitors that target the protease P’ side or its allosteric and orthosteric sites may be better options to counter the flavivirus infection in human individuals. The pharmacokinetic characteristics such as absorption, distribution, metabolism, and excretion properties of these compounds also need to be investigated in order to bring an effective anti-flavivirus drug from the ‘bench to bedside’. The challenging task of flavivirus-specific treatment can be achieved by intensive efforts and sustained multidisciplinary research. The structural similarity between NS2B-NS3 proteases from different members of the Flavivirus family further indicates that studies can be undertaken for designing anti pan flavivirus compounds having broad-spectrum effects against important human flaviviruses like DENV, ZIKV, YFV, WNV, and JEV.
Round 2
Reviewer 2 Report
The authors have done a good job addressing the previous issues raised. I only noted 2 typos that should be corrected-
Line 86 - I think 'lined' should read 'linked'
Line 96 - 'lesions' should read 'lessons'